

# Effects of network topology on the performance of consensus and distributed learning of SVMs using ADMM

Shirin Tavara and Alexander Schliep

Data Science and AI division, Department of Computer Science and Engineering, Chalmers University of Technology and University of Gothenburg, Gothenburg, Sweden

## ABSTRACT

The Alternating Direction Method of Multipliers (ADMM) is a popular and promising distributed framework for solving large-scale machine learning problems. We consider decentralized consensus-based ADMM in which nodes may only communicate with one-hop neighbors. This may cause slow convergence. We investigate the impact of network topology on the performance of an ADMM-based learning of Support Vector Machine using expander, and mean-degree graphs, and additionally some of the common modern network topologies. In particular, we investigate to which degree the expansion property of the network influences the convergence in terms of iterations, training and communication time. We furthermore suggest which topology is preferable. Additionally, we provide an implementation that makes these theoretical advances easily available. The results show that the performance of decentralized ADMM-based learning of SVMs in terms of convergence is improved using graphs with large spectral gaps, higher and homogeneous degrees.

## INTRODUCTION

In recent years, the exponential growth of digital data opened new challenges for traditional serial methods concerning computation and storage, or in short scalability. Distributed optimization methods with centralized or decentralized computation strategies are virtually inevitable to tackle the challenges arising from large-scale machine learning problems. Because of issues such as failure of the central processing unit, communication and synchronization overhead, scalability or privacy the centralized strategies in which the training data is centrally accessible and the distributed agents communicate with a central node may perform poorly (*Yang et al., 2019*). These issues can be resolved by casting the centralized parallel problem into a set of decentralized sub-problems. Decentralized distributed optimization methods in which peer to peer communication is carried out and no central processing unit is involved play an important role in solving large-scale problems. Their importance is amplified in scenarios where data are located on distributed nodes, as in edge computing, or where there are concerns preserving privacy (*Forero, Cano & Giannakis, 2010*).

Corresponding author
Shirin Tavara, tavara@chalmers.se

One effective distributed method is the Alternating Direction Method of Multipliers (ADMM) that offers robustness, stability, distributedly parallelizability and provides convergence guarantees (*Tavara, 2019*). Be that as it may, in specific circumstances, ADMM can still suffer from slow convergence (*Cao et al., 2016*; *França & Bento, 2017*). To resolve this issue, a centralized parallel problem can be cast into multiple decentralized sub-problems with respect to consensus constraints on the classifier parameters. To reach the consensus inter-node communication is required (*Tsianos, Lawlor & Rabbat, 2012*). In the decentralized setting of communication, nodes solve the local optimization problems using only the locally accessible data and they update the local variables after communicating with their one-hop neighboring nodes in order to reach the desired consensus. A one-hop neighbor of a node is a directly connected node in the given network/graph. In a fully distributed and decentralized ADMM, there is no need to exchange training data between agents or nodes. Thus the inter-node communication stays fixed and dependent on the network topology (*Forero, Cano & Giannakis, 2010*). Therefore, the network topology has impact on the convergence rate of consensus-based decentralized ADMM in the context of a specific consensus problem (*França & Bento, 2017*; *Tavara & Schliep, 2018*). This leads to the natural question whether this observation generalizes. On one hand, solving the consensus distributed optimization in a fully connected network, i.e., the complete graph, might lead to faster convergence and obtain the global optimal classifier, while using a low-connected network might lead to slower convergence and a sub-optimal classifier (*Tavara & Schliep, 2018*). On the other hand, the consensus update requires inter-node communication and such communication in a fully-connected network lead to an increase in communication complexity compared to a low-connected network (*França & Bento, 2017*). This raises the question which topology is preferable and what are the trade-offs involved in choosing a topology. We study network properties in terms of connectivity, expansion property, and diameter of graphs. In particular, we focus on the properties as follows; the second smallest eigenvalue of the graph Laplacian for describing the connectivity of a graph, diameter for distinguishing graphs that have the same average degree, but not the same longest shortest path to traverse the graph, and finally expansion property for studying how strong the connectivity of the given graph is and whether the graph is bipartite.

We investigate the impact network topology has on the performance of consensus and distributed learning of Support Vector Machines (SVMs) using decentralized ADMM. SVMs are one of the popular supervised machine learning methods that provides good generalization performance and high accuracy for solving regression and classification problems (*Platt, 1998*). In particular, we focus on answering questions as follows with regards to the ADMM-based SVM algorithm; how much do the expansion property, connectivity and diameter of the network influence the convergence and the training time, how much does the homogeneous degree of a graph affect the convergence and the training time, and which topology is preferable. We also supply an implementation making these advances practically available.

The outline of the paper is as follows. In the next section, we briefly discuss network properties including connectivity, expansion properties, and graph diameter. In the SVMs

and ADMM sections, we give a brief summary of these methods. In the Material and Method section, we describe the algorithm, datasets, network topologies implemented, and other metrics. The results of the experiments and the analysis and discussion of the results are presented in the Results and Discussion section and finally we conclude in the last section.

## Network properties

In this section, we briefly introduce the basics of network topology and properties. In particular, we focus on the connectivity, expansion property, and diameter of graphs.

### Connectivity of graphs

A network of distributed systems can be represented as a connected and undirectional graph $G$ ($V$, $E$), where $V$ is the set of the nodes or agents and $E \subseteq V \times V$ represents connections between the nodes; multiple connection between any two nodes are not allowed. The network topology of a graph $G$ is given by the corresponding adjacency matrix $A(G) = [a_{ij}]_{n \times n}$, where

$$a_{ij} = \begin{cases} 1 & \text{for} (i, j) \in E \\ 0 & \text{otherwise.} \end{cases} \tag{1}$$

The connectivity of a graph is related to the second smallest eigenvalue of the graph Laplacian. The Laplacian matrix of a graph, denoted $L(A)$, is a symmetric matrix and $L(A) = D(A) - A(G)$, where $D(A)$ is a diagonal matrix in which each element of the diagonal shows the degree of corresponding node, and $A(G)$ is the adjacency matrix of the graph. $L(A) = [l_{ij}]_{n \times n}$, where

$$l_{ij} = \begin{cases} -1 & \text{for} (i, j) \in E \ \& \ i \neq j \\ d_i & \text{for } i = j \\ 0 & \text{otherwise.} \end{cases} \tag{2}$$

Here, $d_i$ is the degree of node $i$. The eigenvalues of $L(G)$ satisfy

$$0 = \lambda_1 \leq \lambda_2 \leq \ldots \leq \lambda_n \leq 2d_{max}. \tag{3}$$

Here, $d_{max}$ is the largest degree of all nodes.

The algebraic connectivity, also known as the spectral gap of a graph network, is measured using the second smallest eigenvalue of $L(G)$, denoted $\lambda_2$ (*Donetti, Neri & Muñoz, 2006*). The first smallest eigenvalue of the Laplacian ($\lambda_1 = 0$) is not interesting since it is trivial and only shows whether the graph is connected, but not how well the connectivity of the graph is (*Chow et al., 2016*). A large spectral gap refers to non-modularity of the graph and a small spectral gap relates to a small number of edges needed to be eliminated in order to create a bipartite graph. For details see *Donetti, Neri & Muñoz (2006)*.

### Expansion property

The expansion property of a graph shows how a sufficiently large subset of nodes connects to many nodes. Here, well-connectivity means that in order to make a bipartite graph out

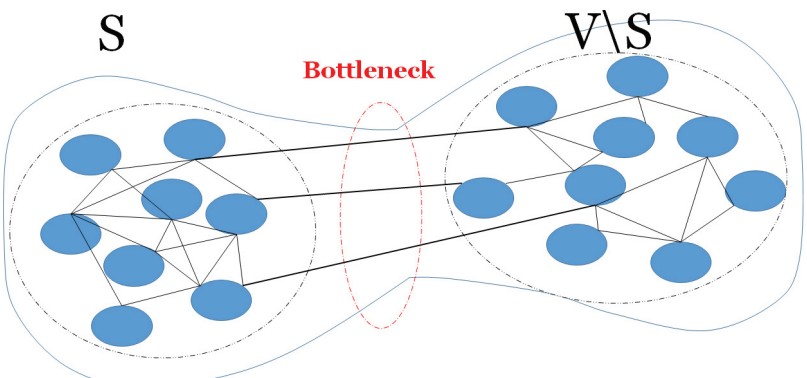

**Figure 1** **A bottleneck on a graph.** An example of a graph with a bottleneck, thus there is few edges (links) between two subsets of vertices.               

of the given graph, a large number of edges should be removed, thus the cut size becomes large. The connectivity of a graph relates to the expansion property of the graph and it can be defined by an isoperimetric or Cheeger constant (*Donetti, Neri & Muñoz, 2006*). The Cheeger constant identifies whether there is a bottleneck in the graph; i.e., whether there are two large subsets of vertices with only few edges between them. A large Cheeger constant shows that there are many edges between any two subsets of vertices, thus no bottleneck. In contrast, a small Cheeger constant shows that the graph has a bottleneck. Figure 1 shows an example of a graph with bottleneck. The Cheeger constant of graph $G(V, E)$, denoted $h(G)$, can be defined as follows,

$$h(G) = min_{S \subseteq V, |S| \leq \frac{|V|}{2}} \frac{|\partial S|}{|S|}. \tag{4}$$

Here, $\partial S = \{(e, e') \in E : e \in S, e' \in V \setminus S\}$. $S$ and $V$ are any two large subsets of vertices. The Cheeger constant is related to the connectivity or spectral gap by Cheeger inequalities, i.e.,

$$\frac{\lambda_2}{2} \leq h(G) \leq \sqrt{2d\lambda_2}. \tag{5}$$

Higher connectivity and spectral gap for a graph can be achieved by enhancing the expansion properties. Graphs that have large Cheeger constants are expander graphs and they are well-known for properties such as sparseness with high connectivity which are preferred properties for constructing efficient and well-connected networks. In this regard, *d*-regular random graphs, in which each node has the same degree *d* and it is connected to *d* other nodes, are expanders if and only if the corresponding spectral gap is lower bounded (*Donetti, Neri & Muñoz, 2006*). In the family of *d*-regular graphs, there is a special type of graph known as the Ramanujan graph which is the optimal expander from the connectivity or the spectral gap point of view (*Parzanchevski, 2018*). A *d*-regular graph is Ramanujan if and only if $|\mu| \leq 2\sqrt{d-1}$ holds for all the eigenvalues, $\mu$, of the

adjacency matrix of the graph. In this paper, we study the impact of Ramanujan graphs on the performance of a parallel distributed ADMM-based SVMs.

## Diameter

Measuring the diameter of graphs helps to distinguish $d$-regular expander graphs from graphs with the same average degree, here, we call the latter mean-degree graphs.

*Klee & Larman (1981)* proved that the diameter of sparse random graphs is *diam* with probability approaching 1 as $n \to \infty$ if,

$$\lim_{n \to \infty} \frac{(pn)^{diam-1}}{n} \to 0 \quad \text{and} \quad \lim_{n \to \infty} \frac{(pn)^{diam}}{n} \to \infty. \tag{6}$$

Here $n$ is the number of vertices and $p$ is the probability of an edge in the graph. Suppose we design $d$-regular graphs or graphs with the average degree of $d$, the graph would have $d \cdot \frac{n}{2}$ edges out of a maximal possible number of edges of the binomial coefficient Bin$(n, 2)$. Hence, the probability of an edge is

$$p = \frac{d \cdot \frac{n}{2}}{\frac{n \cdot (n-1)}{2}} = \frac{d}{n-1}. \tag{7}$$

From Eqs. (6) and (7) we obtain after taking $\lim_{n \to +\infty} \frac{n}{n-1} = 1$

$$\lim_{n \to +\infty} \frac{(pn)^{(diam-1)}}{n} = \lim_{n \to +\infty} \frac{d^{(diam-1)}}{n} \to 0 \tag{8}$$

and

$$\lim_{n \to +\infty} \frac{(pn)^{(diam)}}{n} = \lim_{n \to +\infty} \frac{d^{(diam)}}{n} \to +\infty. \tag{9}$$

Here *diam* and $d$ respectively represent the diameter and the degree of the graph.

An intuition about the connection between diameter and dimension and how this is reflected in Eqs. (8) and (9) can be gained by considering a breadth-first traversal from an arbitrary vertex $v$ to find its antipodal vertex $w$, i.e., the endpoint of the longest shortest path from $v$ to any $w$, and its distance. The diameter is the maximum over all $v$. If all vertices processed add d vertices to the queue, then after *diam* $- 1$ rounds there will be $d^{diam-1}$, an insufficient number to assure that the antipodal vertex is among them per Eq. (8). Note this is asymptotic as $n \to \infty$ and the argument given by Chung and Lu assumes $p$ is fixed which requires $d$ to grow as $n$ increases, therefore it does not directly help giving the diameter for fixed $n$, however it shows the general idea of the effect of an increase in diameter. Expander graphs and in particular $d$-regular graphs are known as highly connected graphs with low diameters. This distinguishes expanders from random graphs with the same mean or average degree.

## SUPPORT VECTOR MACHINES

Support Vector Machines are a supervised machine learning method to solve binary, multi-class classification and regression problems. The original focus of SVMs was on

binary classification to separate two classes of the training samples by finding a hyperplane that has the maximum distance from the closets points on either sides of the hyperplane. The parameters of the hyperplane construct the training model which is used to predict the class label of new and unseen test samples (*Vapnik, 2013*). In order to find a hyperplane with the maximum margin, SVMs solve the optimization problem of the form as follows,

*Primal* :
$$min_{\mathbf{w},b} \quad \frac{1}{2}||\mathbf{w}||^2 + C\sum_{i=1}^{N}\xi_i$$
$$s.t. \quad y_i(\mathbf{w}^T\Phi(\mathbf{x}_i) + b) \geq 1 - \xi_i, \quad i = 1,\ldots,N$$
$$\xi_i \geq 0, \quad i = 1,\ldots,N.$$
(10)

Here, $\mathbf{w}$ is the weight vector of the separating hyperplane, i.e., it is orthogonal to the hyperplane, $\mathbf{x}_i$ is a vector of training samples, $y_i$s are the pre-defined labels of the training samples and $y_i \in \{+1, -1\}$, $b$ is the bias parameter, $\xi_i$s are the classification error and $\Phi(\mathbf{x})$ is the map function. In the linear classification scenario where the training samples are linearly separable, the map function is simply $\Phi(\mathbf{x}) = \mathbf{x}$. The strength of SVMs algorithm appears in real-world scenarios in which the training samples can not always be classified by a linear classifier. In that case, SVMs map training data into a higher dimensional space where data can be linearly separable (*Byun & Lee, 2002*). However, mapping data into a higher dimensional space increases the complexity of the problem known as the curse of dimensionality. To overcome this, SVMs use the kernel trick in which instead of explicitly identifying the mapping function, it is replaced by the product of mapping functions known as kernel functions. Some of the well-studied and -known kernel functions are Gaussian also known as Radial Basis Function (RBF), polynomial and sigmoid kernel functions. In order to use the kernel tricks, the primal optimization problem should be reformulated as the dual optimization problem represented as follows,

*Dual* :
$$min_\alpha \quad D(\alpha) = \frac{1}{2}\alpha^T Q\alpha - \mathbf{1}^T\alpha$$
$$s.t. \quad \sum_{i=1}^{N} y_i\alpha_i = 0$$
$$0 \leq \alpha_i \leq C, \quad i = 1,\ldots,N.$$
(11)

Here, $\alpha$ is Lagrangian multipliers and $\alpha_i \in \alpha$, $\mathbf{1}^T$ is a vector of ones and $Q = [q_{ij}]_{N \times N}$, where $q_{ij} = y_iy_j \Phi^T(\mathbf{x}_i)\Phi(\mathbf{x}_j) = y_iy_j K(\mathbf{x}_i, \mathbf{x}_j)$, $K$ is a kernel function.

## ALTERNATING DIRECTION METHOD OF MULTIPLIERS

Distributed optimization methods are important approaches for solving large-scale machine learning problems in a distributed manner. In these methods, the global optimization problem which is formed as the sum of the local optimization problems is solved by the cooperation of multi-agent systems. Each agent solves the local optimization problem using its own local data and thereafter performs the updates based on the

 

information received from neighboring agents through underlying communication network (*Yang et al., 2019*). One of the major application is in edge computing in which each device independently performs it own local computation and communicates with other devices through a communication network.

Alternating Direction Method Of Multipliers (ADMM) is one of the powerful approaches for solving an optimization problem in a distributed manner. The popularity of ADMM comes from properties such as decomposability, the adaptability to solve distributed optimizations in a decentralized manner, robustness, scalability and parallelizability. Besides, the strong convergence property of ADMM regarding convex functions along with no requirement of diffrerentioability of the objective function describe the robustness of this method (*França & Bento, 2017*). For convex functions, which is the case in SVMs, the convergence rate of ADMM is proven to be $O(1/N)$, where $N$ is the number of ADMM interations (*França & Bento, 2017*; *He & Yuan, 2012*; *Monteiro & Svaiter, 2013*). In practice the effect of network topology on the convergence and the convergence rate of ADMM are poorly understood (*França & Bento, 2017*; *Ghadimi et al., 2012*). The general optimization problem

$$min_{\mathbf{w}} \quad f(\mathbf{w}) \tag{12}$$

can be transformed into a consensus optimization problem of the form Eq. (13) and it can be solved in a distributed and decentralized manner in which each agent performs the computation using the local data to obtain the optimal solution $\mathbf{w}$. The process continues until it reaches the global consensus $\mathbf{w}$. In the global consensus and centralized model, all agents communicate with a central or a master agent to perform the update of their local computations. Thus, we arrive at the distributed optimization equivalent to Eq. (12), i.e.,

$$min_{\mathbf{w}_j} \quad \sum_{j=1}^{N} f_j(\mathbf{w}_j) \tag{13}$$
$$s.t. \quad \mathbf{w}_j - \mathbf{w} = 0, \quad j = 1, \ldots, N.$$

Here, $\mathbf{w}_j$s are the local solution of the system and $\mathbf{w}$ is the global consensus variable. The centralized distributed optimization in Eq. (13) can be solved in a decentralized manner shown in Eq. (14) in which each agent performs local updates by only communicating with neighboring agents through the underlying communication network. The process continues until each agent reaches the consensus, i.e., the local optimization solution gets close enough to the solutions of neighboring agents.

$$min_{\mathbf{w}_j} \quad \sum_{j=1}^{N} f_j(\mathbf{w}_j) \tag{14}$$
$$s.t. \quad \mathbf{w}_j - \mathbf{w}_i = 0, \quad j = 1, \ldots, N, i \in \mathcal{N}_j.$$

Here, $\mathcal{N}_j$ is the one-hop neighbor nodes of node $j$. In the constraints defined in Eq. (14), local variables $\mathbf{w}_j$s are forced to agree across one-hop neighbors $\mathbf{w}_i$s. For detailed information, refer to *Forero, Cano & Giannakis (2010)* and *Boyd et al. (2011)*.

## Training SVMs using ADMM

To make the exposition self-contained, we summarize some crucial steps on how ADMM is applied to the training of SVMs. *Forero, Cano & Giannakis (2010)* introduced the consensus and decentralized reformulation of Eq. (10) as

$$
\begin{aligned}
&min_{\mathbf{w}_j, b_j, \xi_j} && \frac{1}{2}\sum_{j=1}^{J}||\mathbf{w}_j||^2 + JC\sum_{j=1}^{J}\mathbf{1}_j^T\xi_j \\
&s.t. && \mathbf{Y}_j(\Phi(\mathbf{X}_j)\mathbf{w}_j + \mathbf{1}_j b_j) \geq \mathbf{1}_j - \xi_j, && j = 1, ..., J \\
& && \xi_j \geq \mathbf{0}_j, && j = 1, ..., J \\
& && \mathbf{G}\mathbf{w}_j = \mathbf{G}\mathbf{w}_i, && j = 1, ..., J, i \in \mathcal{N}_j \\
& && b_j = b_i, && j = 1, ..., J, i \in \mathcal{N}_j.
\end{aligned}
\tag{15}
$$

Here, $\mathbf{X}_j = [\mathbf{x}_{j1}, ..., \mathbf{x}_{jNj}]^T$ and $\mathbf{Y}_j = \text{diag}([y_{j1}, ..., y_{jNj}])$. To guarantee the convergence, the optimization is performed over a connected network with $J$ nodes represented by an undirected graph with vertices $\{1, ..., J\}$. The last two constraints in Eq. (15) are the consensus constraints forcing local variables $\mathbf{w}_j$s and $b_j$s to agree with variables across the one-hop neighbor nodes, i.e., $\mathbf{w}_i$ and $b_i$. Applying ADMM on the non-linear SVM classification requires the ADMM updates to be carried out in the Hilbert space also known as the feature space (*Rossi & Villa, 2005*). The dimension of the feature space in which the data can be linearly separable may increase to infinity. Thus the computation and communication complexities are increased. As we mentioned in section Support Vector Machines, the phenomenon is known as the curse of dimensionality. To reduce the complexities, *Forero, Cano & Giannakis (2010)* project the consensus constraints into a reduced rank subspace using matrix $\mathbf{G}$. Here, $\mathbf{G} = [\Phi(\chi_1), ..., \Phi(\chi_L)]^T$ with predefined vectors $\{\chi_l\}_{l=1}^{L}$. The size and the choice of these vectors determine the similarity between the local classifiers. A large $L$ leads to similar classifiers with the price of higher computation and communication complexities. In contrast, a small $L$ contributes to reduced complexities with the price of dissimilar local classifiers. We consider a fixed $L$ for all the network topologies. For details about determining appropriate $L$ see *Forero, Cano & Giannakis (2010)*. Applying ADMM to solve Eq. (15) leads to the following updates,

$$
\{\mathbf{w}_j(t+1), b_j(t+1), \xi_j(t+1)\} = \text{argmin}_{\{\mathbf{w}_j, b_j, \xi_j\}}\mathcal{L}'(\{\mathbf{w}_j\}, \{b_j\}, \{\xi_j\}, \{\alpha_j(t)\}, \{\beta_j(t)\})
\tag{16}
$$

$$
\alpha_j(t+1) = \alpha_j(t) + \frac{\eta}{2}\sum_{i \in \mathcal{N}_j}\mathbf{G}[\mathbf{w}_j(t+1) - \mathbf{w}_i(t+1)]
\tag{17}
$$

$$
\beta_j(t+1) = \beta_j(t) + \frac{\eta}{2}\sum_{i \in \mathcal{N}_j}[b_j(t+1) - b_i(t+1)].
\tag{18}
$$

Here, $\mathcal{L}'$ is the corresponding augmented Lagrangian function and $\eta > 0$ is the corresponding ADMM constant. The interesting point is that there is no need to explicitly solve the ADMM iterations Eqs. (16)–(18) since $\mathbf{w}_{j(t)}$s are located in the high-dimensional feature space. Besides the local discriminant functions can be determined using the SVM kernels without explicitly finding $\mathbf{w}_{j(t)}$s. This is shown in Eq. (24). To do this, the iteration Eq. (16) is solved by its dual Eq. (19). Suppose $T = [\chi_1, ..., \chi_L]^T$ with predefined

vectors $\{\chi_l\}_{l=1}^L$, the kernel matrices $\mathbf{K}(.,.)$ and $\tilde{K}(.,.)$ and $\tilde{\mathbf{w}}_j(t) = \mathbf{G}\mathbf{w}_j(t)$, *Forero, Cano & Giannakis (2010)* transformed the ADMM iterations Eqs. (16)–(18) into

$$\boldsymbol{\lambda}_j(t+1) = argmax_{\boldsymbol{\lambda}_j : \mathbf{0}_j \preceq \boldsymbol{\lambda}_j \preceq JC\mathbf{1}_j} - 0.5\boldsymbol{\lambda}_j^T \mathbf{Y}_j(\mathbf{K}(\mathbf{X_j},\mathbf{X_j}) - \tilde{\mathbf{K}}(\mathbf{X_j},\mathbf{X_j}) + \frac{\mathbf{1}_j\mathbf{1}_j^T}{2\eta|\beta_j|})\mathbf{Y}_j\boldsymbol{\lambda}_j$$

$$+ \mathbf{1}_j^T\boldsymbol{\lambda}_j - (\tilde{\mathbf{f}}_j^T(t)(\mathbf{K}(\mathbf{T},\mathbf{X_j}) - \tilde{\mathbf{K}}(\mathbf{T},\mathbf{X_j}) + h_j(t)\frac{\mathbf{1}^T}{2\eta|\beta_j|})\mathbf{Y}_j\boldsymbol{\lambda}_j, \tag{19}$$

$$\tilde{\mathbf{w}}_j(t+1) = [\mathbf{K}(\mathbf{T},\mathbf{X_j}) - \tilde{\mathbf{K}}(\mathbf{T},\mathbf{X_j})]\mathbf{Y}_j\boldsymbol{\lambda}_j(t+1) - [\mathbf{K}(\mathbf{T},\mathbf{T}) - \tilde{\mathbf{K}}(\mathbf{T},\mathbf{T})]\tilde{\mathbf{f}}_j(t), \tag{20}$$

$$b_j(t+1) = \frac{1}{2\eta|\beta_j|}[\mathbf{1}_j^T\mathbf{Y}_j\boldsymbol{\lambda}_j(t+1) - h_j(t)], \tag{21}$$

$$\alpha_j(t+1) = \alpha_j(t) + \frac{\eta}{2}\sum_{i\in\mathcal{N}_j}[\tilde{\mathbf{w}}_j(t+1) - \tilde{\mathbf{w}}_i(t+1)], \tag{22}$$

$$\beta_j(t+1) = \beta_j(t) + \frac{\eta}{2}\sum_{i\in\mathcal{N}_j}[b_j(t+1) - b_i(t+1)]. \tag{23}$$

Here, $\boldsymbol{\lambda}_j$s are the Lagrange multipliers. Consequently the local discriminant functions are computed using the kernel trick as follows,

$$g_j^{(t)}(\mathbf{x}) = \sum_{n=1}^{N_j} a_{jn}(t)K(\mathbf{x},\mathbf{x}_{jn}) + \sum_{l=1}^{L} c_{jl}(t)K(\mathbf{x},\chi_l) + b_j(t). \tag{24}$$

For details about the theoretical analysis and the convergence proof see *Forero, Cano & Giannakis (2010)* and the provided appendices.

# MATERIAL AND METHOD

## Algorithm

The ADMM-based SVMs algorithm, algorithm 3, given by *Forero, Cano & Giannakis (2010)* served as a base of our C++ implementation for solving non-linear classification problems in this paper. The iterations and the communication process in the algorithm are summarized and illustrated in Fig. 2. As shown in subplot (A), every node $j$ solves the local optimization problem and computes $\lambda_j(t+1)$ which is used to obtain $V_j(t+1)$ and then $V_j(t+1)$ is broadcasted to all the one-hop neighboring nodes, here we used augmented vector $V_j = [\mathbf{w}_j^T, b_j]^T$ and the neighboring nodes of node $j$ are $\{j_1, j_2, j_3, j_4\}$. As shown in subplot (B), once every node $j$ receives and gathers the corresponding $V$ from all the neighbors, it computes $\alpha_j(t+1)$. The advantage of the aforementioned communication is that in the broadcasting phase, every node $j$ sends a message of fixed size $L$ to the neighboring nodes per iteration, here $L$ is the size of $V_j$. Similarly in the gathering phase, every node receives messages of the same size from the neighbors. Note $\lambda_j(t+1)$ and $\alpha_j(t+1)$ are not exchanged among the nodes.

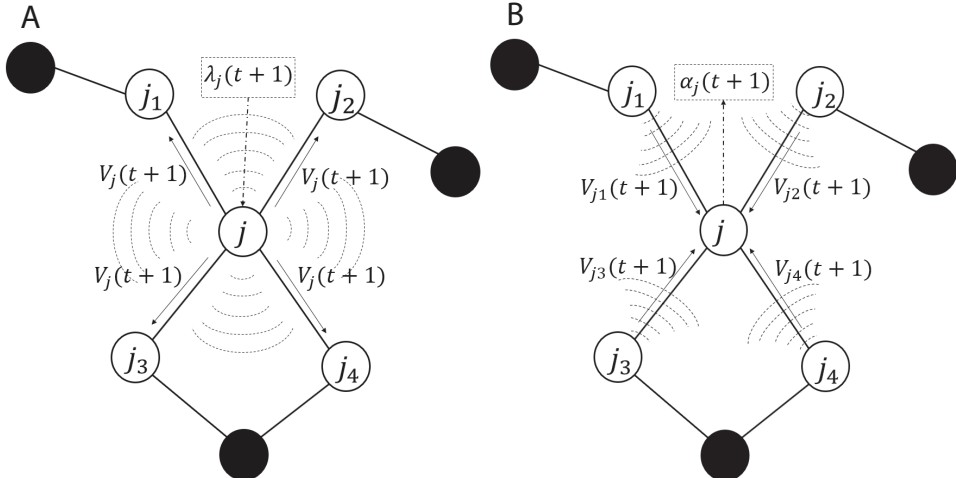

**Figure 2 Communication including the (A) broadcasting and (B) gathering in the ADMM-based SVMs.** White circles are one-hop neighbors to node j and black circles are non-neighbors.

To the best of our knowledge this algorithm is the state-of-the-art ADMM with decentralized and distributed learning of SVMs. Our implementation can handle solving linear and non-linear classification problems with the focus on decentralized and distributed computing to reach consensus.

## Datasets

To conduct numerical experiments, we used a biological dataset, micro RNAs precursors (`pre-miRNAs`) gathered by *Lopes, Schliep & De Carvalho (2016)*. They created the dataset by extracting 85 features describing sequence and structural aspects of `pre-miRNAs` sequences (positive class) and hairpin like sequences (negative class). The other datasets are from public data repositories gathered in LIBSVM Data (*Chang & Lin, 2011*) and UCI (*Dua & Graff, 2017*). Table 1 shows the datasets used in the experiments and the corresponding number of training instances and features. The format of the datasets that we used is of the format of LIBSVM. Here, we briefly describe each dataset. `Susy` comprises two classes, one for signal processes which produce supersymmetric particles, and one for background processes which do not. The dataset has 18 distinct features or attributes, the first eight features represent kinematic properties measured by particle detectors. The next 10 features are functions of the first features derived by physicists to discriminate between the two classes (*Baldi, Sadowski & Whiteson, 2014*). `Higgs` comprises two classes, signal processes that produce `Higgs` bosons and background processes that do not. From 28 kinematic features, the first 21 features represent properties measured by particle detectors. The last 7 features are the functions of the first 21 features derived by physicists to discriminate between two classes (*Baldi, Sadowski & Whiteson, 2014*). `Covtype` is used to predict the forest cover type from cartographic variables. A total of 54 features in the dataset consisted of elevation, aspect, slope and other information

**Table 1 Dataset information.**

| Datasets | Training points | Features |
|---|---|---|
| Susy | 700,000 | 18 |
| Higgs | 500,000 | 28 |
| Covtype | 480,960 | 54 |
| SkinNonSkin | 346,800 | 3 |
| Cod-rna | 180,960 | 8 |
| Seismic | 97,080 | 50 |
| Pre-miRNAs | 16,080 | 85 |

derived from spatial data obtained from geographic information system and US forest service information (*Dua & Graff, 2017*; *Blackard & Dean, 1999*). SkinNonSkin is skin segmentation dataset consisted of skin and non-skin classes. The dataset is generated from skin textures of face images. Three features in the dataset represent age groups (young, middle and old), race groups (white, black and Asian), and genders (*Dua & Graff, 2017*; *Bhatt & Dhall, 2012*). Cod-rna is used to find non-coding RNAs in sequenced genomes to understand/discover candidate drug targets. The features are related to a common structure of two RNA sequences. The first feature represents the value computed by the Dynalign algorithm, the second feature is the length of the shorter sequence, and the rest represents nucleotide and dinucleotide frequencies of the sequences (*Uzilov, Keegan & Mathews, 2006*). Seismic is used for vehicle type classification using seismic signals sensed by the sensors. The features are extracted from the sensor data including time series data observed at each sensor and sensor's microphones (*Duarte & Hu, 2004*).

## Network topology

We constructed four groups of graphs. The first group comprises random $d$-regular expander graphs, in particular Ramanujan graphs. We followed the conditions described in the Expansion Property section to construct Ramanujan graphs. The second group of the graphs consists of random graphs with $d$-mean degree, i.e., each node of the graph connects in average to the $d$ number of nodes. Note in this group of the graphs, the degree of nodes may not be the same. The third group consists of the complete graphs in which each node has degree $N - 1$ and $N$ is the number of graph nodes. Finally, the fourth group consists of line, ring, 2-D torus, and star graphs representing some of the common modern network topologies. We constructed aforementioned type of the graphs with different degrees and/or number of graph nodes depending on the size of training instances. For training datasets with a smaller number of instances, we generated graphs with low degrees and for those of with larger instances, we created graphs with higher degrees.

## Binary classification and parallel computation

In this article, we focused on binary classifications since multi-class classifications can be transformed into several binary classifications using the one-vs.-all or the one-vs.-one

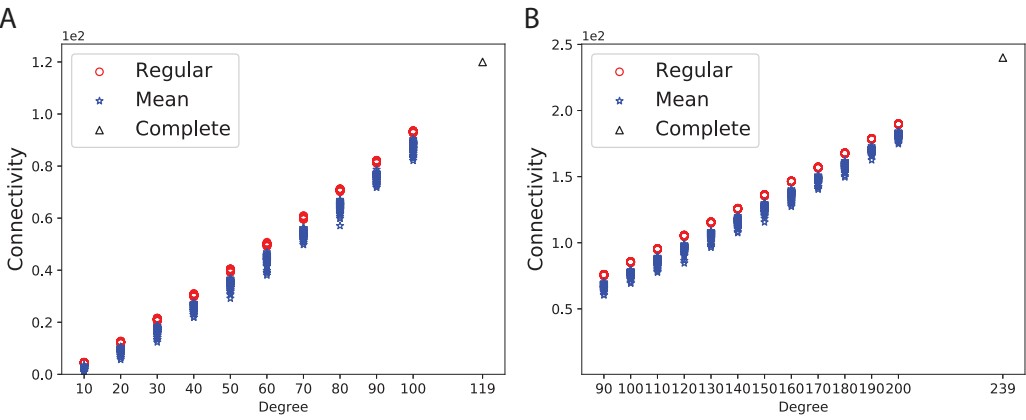

**Figure 3 Connectivity of the 100 regular, 100 mean-degree and complete graphs.** (A) for 120 graph nodes and (B) for 240 graph nodes.

techniques. We used the Message Passing Interface (MPI) (*Barker, 2015*) for communication between nodes and mapped the graph nodes into the distributed cores.

## Parameter tuning

The classification accuracy of a distributed ADMM-based SVM is influenced by important parameters of the algorithm, i.e., $\gamma$, $J$, $C$ and $\rho$. $\gamma$ is the parameter of RBF kernel, $J$ is the number of nodes/agents, $C$ is the regularizer that has the duty of balancing the misclassification error, and $\rho$ is the ADMM parameter. We tune the parameters using a grid search with cross-validation. In case we get poor accuracy, we apply standardizing and normalizing techniques.

## Evaluation metrics

We evaluate the results using the standard statistics, i.e., accuracy metrics, true positive/negative rate, and positive/negative predictive rate (*Tharwat, 2020*). To better understand the impact of network topology on the performance of the algorithm, we measure iterations, total time, and scaled communication time during training. Besides, we measure total time, maximum iterations of the inner optimization solver, and time versus errors for mean-degree and regular graphs.

## RESULTS AND DISCUSSION

In this section, we briefly describe the findings and insights from the results. We highlight some of the results and a more comprehensive view is presented in the supplement.

## Connectivity and expansion properties

To obtain a better insight into the graph connectivity, we constructed 100 expander (Ramanujan) and mean-degree graphs with the given degrees and the number of nodes. As a base-line, we constructed complete graphs. Figure 3A shows the connectivity of 100 random $d$-regular and mean-degree graphs for $N = 120$ and $d = \{10, 20, 30, 40, 50, 60, 70, 80, 90, 100\}$ along with the corresponding complete graph. Figure 3B shows the connectivity of the same types of graphs for $N = 240$ and $d = \{90, 100, 110, 120, 130, 140,$

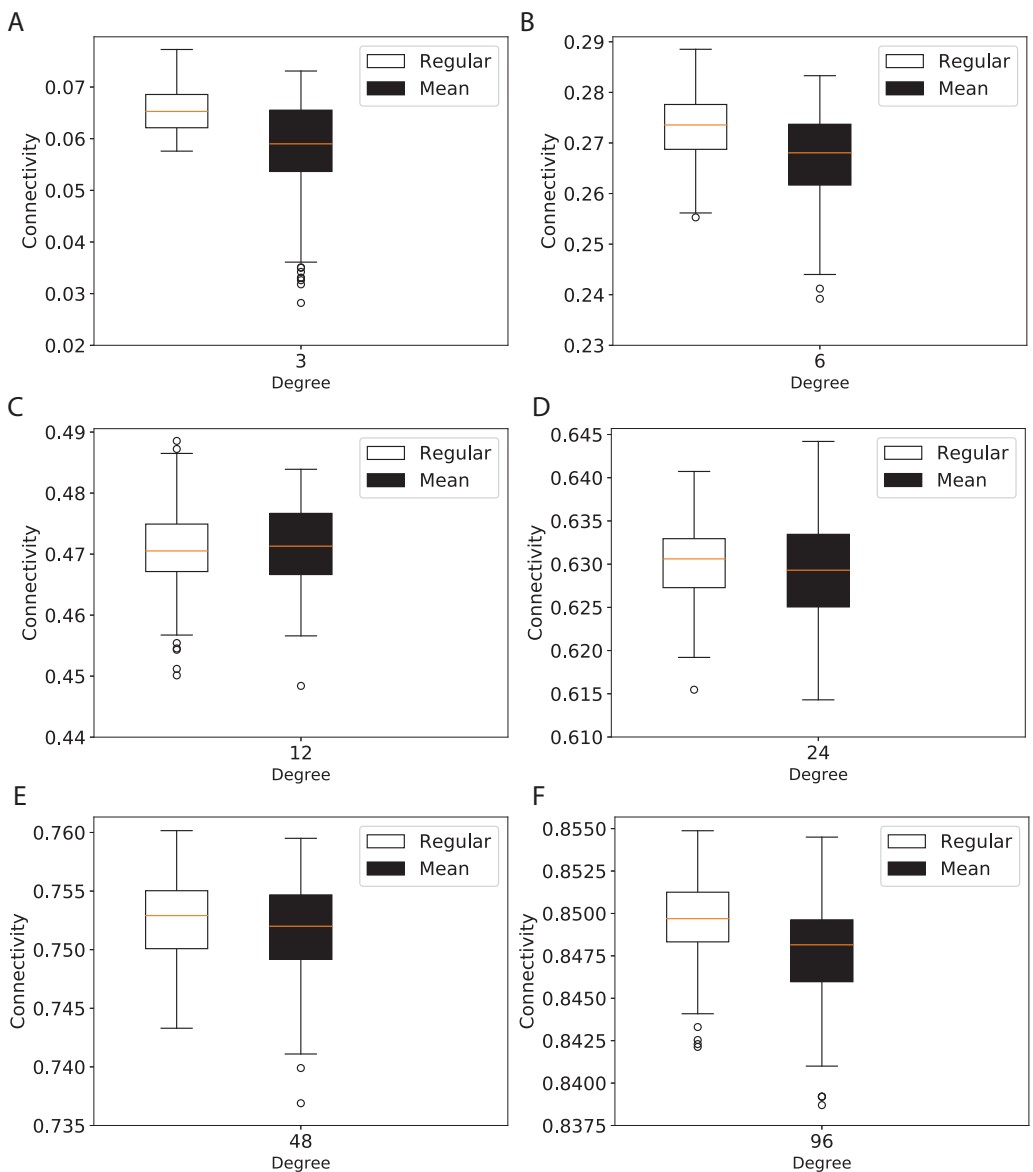

**Figure 4 Connectivity of the 100 regular and 100 mean-degree graphs with 240 graph nodes.** (A) for degree 3, (B) for degree 6, (C) for degree 12, (D) for degree 24, (E) for degree 48, and (F) for degree 96.

150, 160, 170, 180, 190, 200}. As Figs. 3A and 3B show, the connectivity of mean-degree graphs is similar or lower than the connectivity of the regular graphs for the same degree and the number of nodes, however the variation of connectivity is higher in the mean-degree graphs than in the regular graphs. As expected, the complete graphs in both figures have the highest connectivity.

To study the connectivity for networks similar to the modern network topologies, we constructed expander and mean-degree graphs for lower degrees $d = \{3, 6, 12, 24, 48, 96\}$. For a fair comparison with a similar scale for the connectivity, we used the second smallest eigenvalue of the normalized Laplacian matrix. Figure 4 shows the box plots

for the connectivity of 100 graphs constructed for the regular and mean-degree groups. In each sub-plot, the upper and lower horizontal lines respectively represent the maximum and minimum connectivity obtained for 100 graphs. The orange horizontal lines in the body of the sub-plots represent the median of the connectivity regarding the regular and mean-degree graphs constructed. It is visible that, the median and minimum connectivity of mean-degree graphs are lower than that of the regular graphs. The maximum connectivity of mean-degree graphs, except for degree 24, is lower than that of the regular graphs. Besides, the variation in the connectivity for the mean-degree graphs is higher than that of the regular graphs. This suggests that the connectivity of a mean-degree graph will with a higher probability be lower than that of a random regular/expander graph with the same average degree. These observations are more visible for graphs with lower degrees, i.e., $d = \{3, 6, 12\}$. For the higher degrees, i.e., $d = \{24, 48, 96\}$, the variation in the connectivity is reduced. Hence, the connectivity of mean-degree and regular graphs gets closer as the degree increases towards the complete graphs.

## Diameter

We complement our analysis by exploring the diameters of the different networks constructed. We generated 5,000 random $d$-regular and $d$-mean degree graphs for $d = \{3, 6, 12, 24, 48, 96\}$, for which we computed the diameters. Results from prior work by *Ludu (2016)* suggest that the diameters of the random regular graphs can be approximated by

$$diam(G) \sim log_{d-1}(n) + log_{d-1}(ln(n)). \tag{25}$$

Here $n$ is the number of vertices and $d$ is the degree of graph $G$. Our results confirm roughly comparable diameters approximation for 5000 random regular graphs, i.e., $diam(G) \pm 1$. However, the diameter for random mean-degree graphs varies a lot more than those for random regular graphs, i.e., $2 \times (diam(G) \pm 1)$. In the Diameter section, page 4, we discussed the general idea of the effect of an increase in diameter. Based on the results, diameters of mean-degree graphs are worse than those of regular graphs. This might suggest that the inhomogeneity of graph degrees may impact the longest shortest path of the graph. The difference between the diameters of regular and mean-degree graphs decreases as the connectivity and the degree of nodes increase from 96 all the way towards complete graphs.

The diameters and the connectivity of the random mean-degree graphs are shown in Fig. 5. From Fig. 5A, it is visible that the variation of diameters for lower degree graphs, $d = \{3, 6\}$ is higher than those for the higher degree graphs, $d = \{12, 24, 48, 96\}$. This might suggest that the diameter of a random mean-degree graph for $d = \{3, 6\}$ has a lower probability to reach its close-to-optimal diameter than that of a higher mean-degree graph. The optimal diameter is the diameter with the lowest possible value. The diameter improves as the degree of graphs increases. The variation in diameters dampens as the degree of graphs increases. The same trend is valid for the connectivity in Fig. 5B, i.e., the variation of connectivity for lower degree graphs is higher than that for the higher degree graphs. As expected, the connectivity enhances as the degree of graphs increases. In general, this suggests that for lower degrees, a random mean-degree

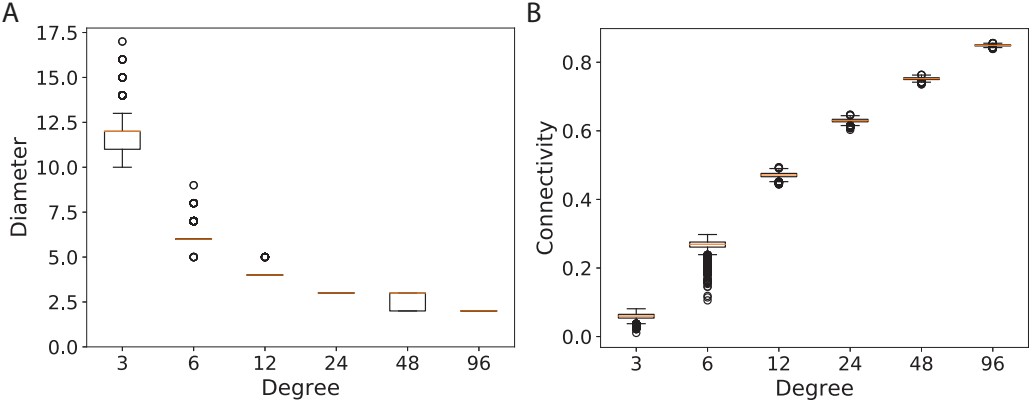

**Figure 5 Diameters and connectivity of 5,000 $d$-mean-degree graphs generated using 240 graph nodes and $d$ = {3, 6, 12, 24, 48, 96}.** (A) The diameters of the graphs and (B) the connectivity of the graphs.

graph has a lower probability to reach a comparable connectivity and diameter of a random regular graph given the same average degree.

## Numerical experiments

To study the impact of network topology on the performance of ADMM-based SVM, we randomly chose $d$-regular and $d$-mean-degree graphs constructed for the given degrees and nodes. Graphs nodes are mapped into distributed computing nodes on the Chalmers cluster, Hebbe. "The Hebbe cluster is built on Intel 2650v3 CPU's. The system consists of in total 315 compute nodes (total of 6,300 cores) with 26 TiB of RAM and 6 GPUs" (*C3SESupport, 2020*).

### *Effect of connectivity increase*

We studied the effect of increase of connectivity on the performance of the algorithm using $d$-regular and complete graphs. In the Connectivity and Expansion Properties section, page 9, we discussed that increase in degree leads to higher connectivity. Here, the results confirm whether it holds and what the corresponding consequences are. Figure 6 shows the number of iterations, training time and communication time per iteration for `Susy`, `Higgs`, `Covtype` and `SkinNonSkin` datasets using regular and complete graphs. Here, 240 graph nodes are mapped into 240 computing cores and $d$ = {100, 110, 120, 130, 140, 150, 160, 170, 180, 190, 200, 210, 239 }.

The results shown by subplots (A) in Fig. 6 and those in the supplement suggest a similar trend for the regular and complete graphs and that is that the number of iterations decreases as the degree of graphs becomes larger. The most visible part of the trend for each dataset is as follows; for `Susy` between 110 and 180 degree, for `Higgs` between 100 and 190, for `Covtype` between 100 and 210 degree, and for `SkinNonSkin` between 110 and 140 degree. Note the trend of decreasing the number of iterations saturates as the degree of the nodes gets closer to the total number of graph node; i.e., the graph gets closer to being the complete graph. In some cases, the number of iterations until convergence behaves adversely and increases for the complete graphs due to the need to reach

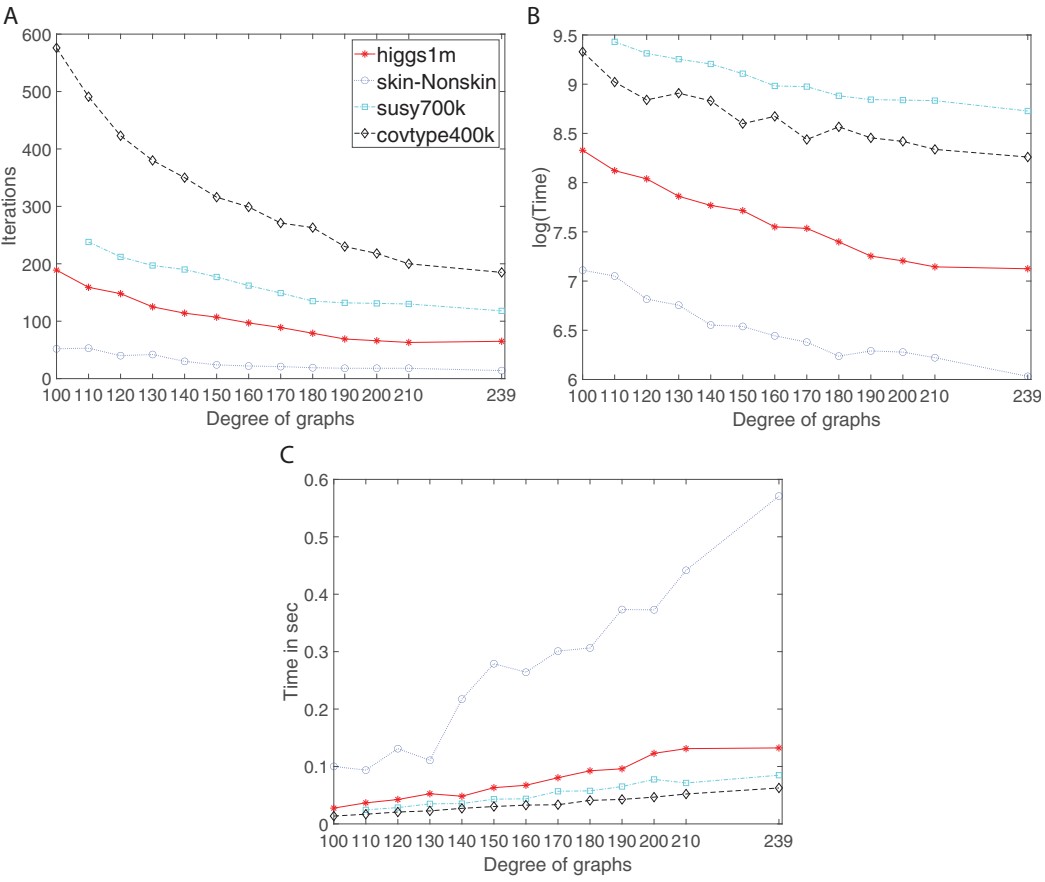

**Figure 6 Iterations, total training time, and scaled communication time for `Higgs`, `SkinNonSkin`, `Susy` and `Covtype` using regular and complete graphs.** (A) Iterations for different degrees, (B) Training time for different degrees and (C) Scaled communication time during training.

consensus between many nodes. For instance, this happens for `Higgs` in Fig. 6A and for `Seismic` in Fig. 2A in the supplement. Subplots (B) in Fig. 6 and all the figures in the supplement represent the training time in which the training gets faster as the degree of graphs becomes larger, i.e., the ADMM-based SVMs require fewer iterations for the higher degree of graphs to reach the target or given accuracy. This is most noticeable between 100 and 190 degree for `Higgs`, between 110 and 180 for `SkinNonSkin`, between 110 and 160 for `Susy` and between 100 and 150 for `Covtype`. As the degrees get closer to the number of nodes the training time eventually plateaus. This is due to the increasing communication time since many nodes need to reach consensus as the degrees increase. This is shown in subplots (C) in Fig. 6 and all the figures in the supplement in which the communication time increases as the degree of graphs becomes larger. This is most noticeable for `SkinNonSkin` between 110 and 239 degree, for `Higgs` between 100 and 200 degree, for `Susy` between 100 and 200, and for `Covtype` between 100 and 239. The results shown by subplots (C) suggest that the communication overhead increases as the graph gets closer to being the complete graph even though the number of iterations decreases.

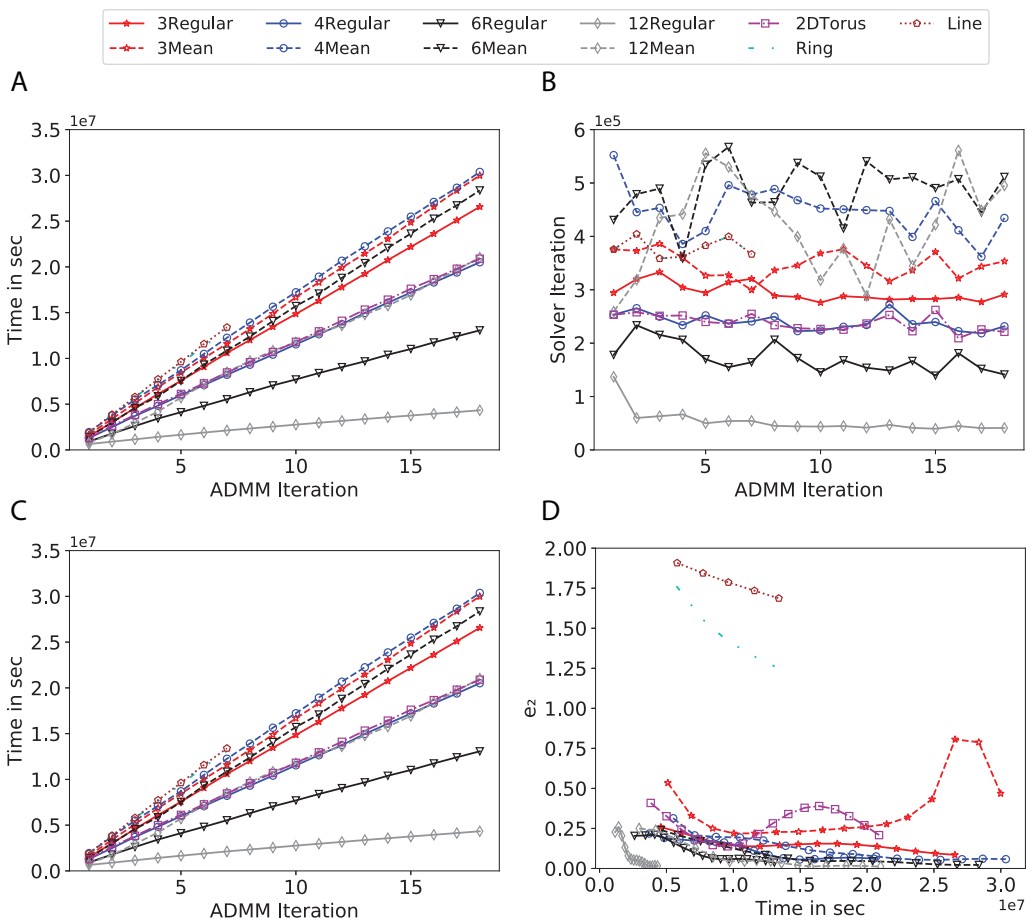

**Figure 7** **Total time, maximum iterations of the inner optimization solver, and time vs. e1 and e2 for** `Higgs` **using line, ring, 2DTorus,** *d*-**mean-degree and d-regular graphs where** *d* = **{3, 4, 6, 12}.**
(A) Total elapsed time of the training phase, (B) Maximum iteration in the inner optimization solver,
(C) Difference between the result of node j and its neighbors, time vs. e1 and (D) Difference between the
result of node j and the average results, time vs. e2.     

### Effect of homogeneous degree distribution

To study the effect of homogeneous degrees of graphs/networks on the performance of
ADMM-based SVMs, we trained the datasets using mean-degree graphs and compared
the results to regular graphs with the same given degrees and some of the common modern
network topologies represented with line, ring, star, and 2-D torus graphs. We study
the total training time, maximum number of inner solver iterations to solve the
optimization problem at each iteration of ADMM, the elapsed time vs. the difference
between local and neighboring classifiers, and finally the elapsed time vs. the difference
between the local classifier and the average of all classifiers for all type of graphs
constructed. Here, the degrees for mean-degree and random regular graphs are $d$ = {3, 4, 6,
12, 24, 48, 96}. We split the plots of the result for some of the degrees due to dissimilarity
in the scales of y-axis. The results are as follows; Figs. 7–9 respectively represent
the results for `Higgs`, `Covtype` and `Susy` regarding line, ring, star, 2-D torus graphs,
mean-degree and regular graphs with degree $d$ = {3, 4, 6, 12}. Figures 10 and 11

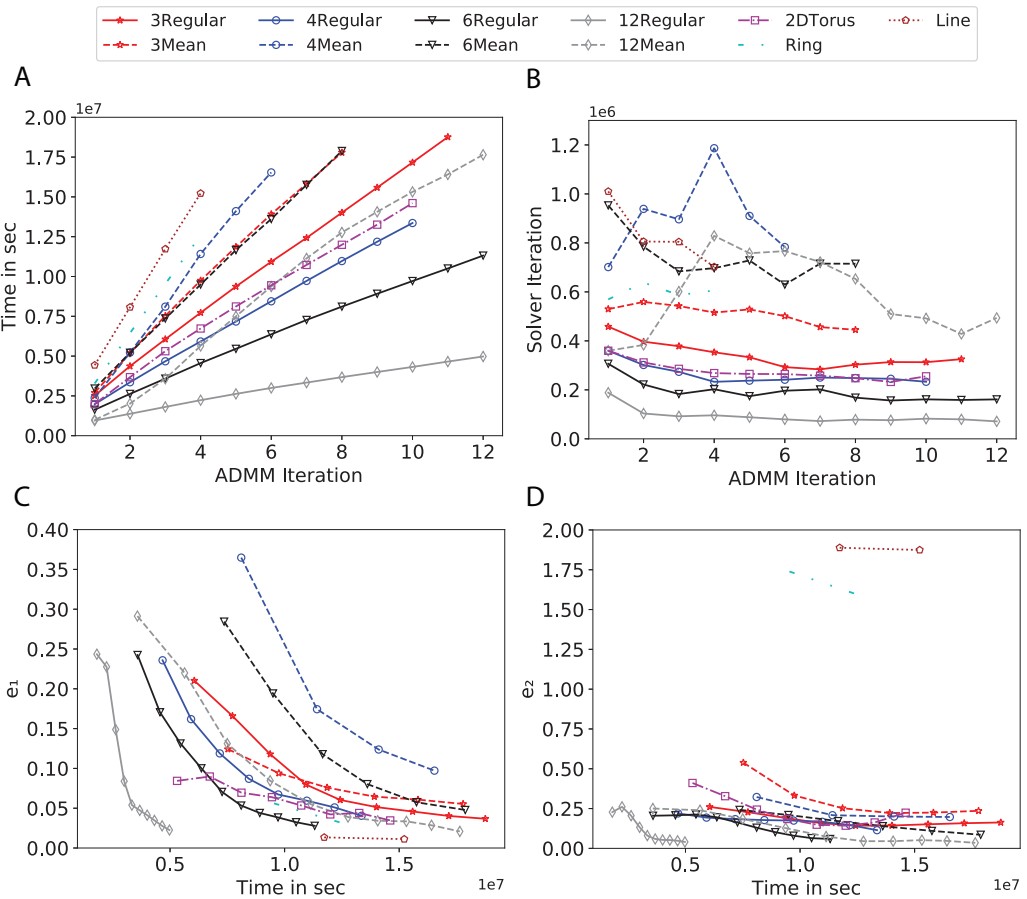

**Figure 8** **Total time, maximum iterations of the inner optimization solver, and time vs. e1 and e2 for** **Covtype** **using line, ring, 2DTorus, *d*-mean-degree and *d*-regular graphs where *d* = {3,4,6,12}.** (A) Total elapsed time of the training phase, (B) Maximum iteration in the inner optimization solver, (C) Difference between the result of node j and its neighbors, time vs. e1 and (D) Difference between the result of node j and the average results, time vs. e2.

respectivelty represent the results for Higgs and Covtype regarding mean-degree and regular graphs with $d = \{24, 48, 96\}$. Note, in some cases, we only consider a limited number of ADMM iterations due to slow training pace and to save CPU hours on the distributed cluster, i.e., we choose 7 ADMM iterations for Higgs using line and ring graphs and 8 ADMM iterations for Covtype using 3 and 6 mean-degree graphs and 4 iterations for line and ring graphs.

Subplots (A) in Figs. 7–11 show that the total training time increases as we increase the number of ADMM iterations for all the graphs constructed. However the gap between the total training time of regular and mean-degree graphs in most cases is large. In subplots 7A–9A, except for degree 3, the gap is large for all the degrees shown. In Figs. 10A and 11A, the most visible gap belongs to degree 48 and 96. In subplots 7A–9, the total training time regarding 2-D torus graph is worse or similar to that of the 4-regular graph and that is better than in the 4-mean-degree graph. Results of subplot (A) suggest that the homogeneity of degrees affects the total training time. Note the small gaps for some of the

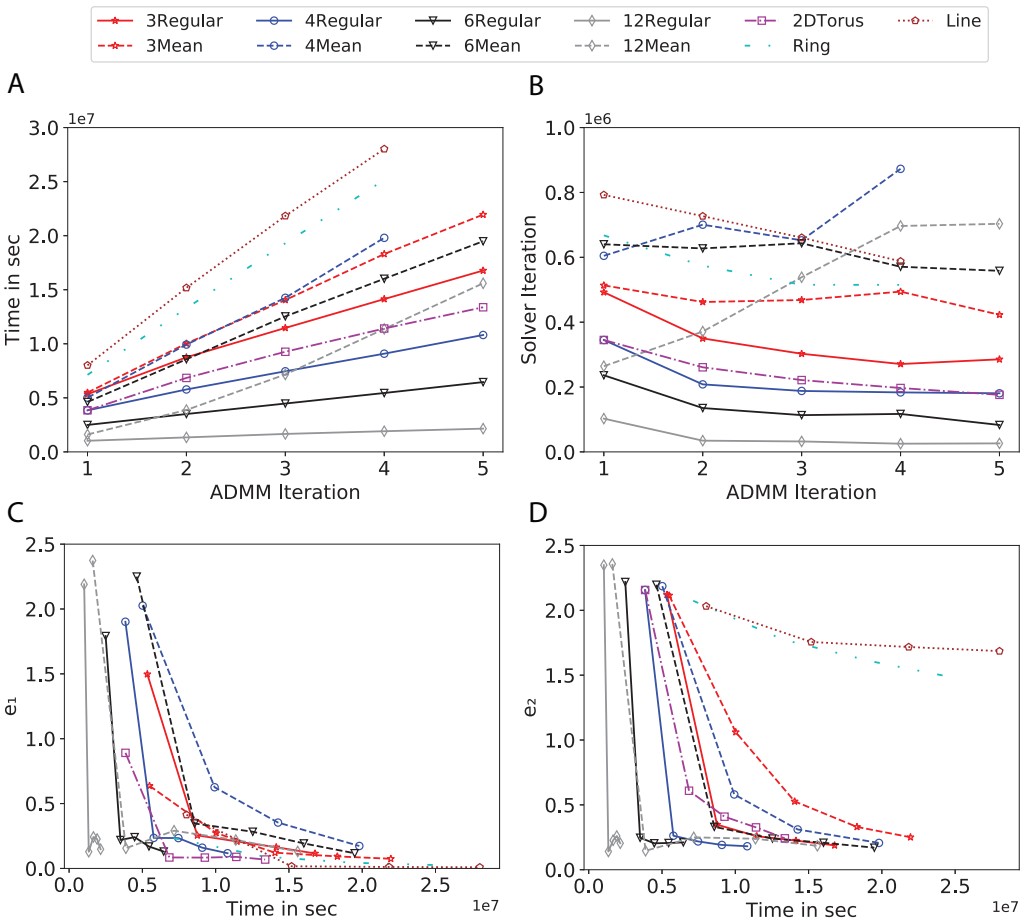

**Figure 9 Total time, maximum iterations of the inner optimization solver, and time vs. e1 and e2 for Susy using line, ring, 2DTorus, *d*-mean-degree and *d*-regular graphs where *d* = {3, 4, 6, 12}.** (A) Total elapsed time of the training phase, (B) Maximum iteration in the inner optimization solver, (C) Difference between the result of node j and its neighbors, time vs. e1 and (D) Difference between the result of node j and the average results, time vs. e2.

degrees in subplots (A) can be due to the similarity between the connectivity of the random regular and mean-degree graphs constructed since the graphs are randomly chosen.

Subplots (B) in Figs. 7–11 represent the maximum number of iterations from solving local SVMs sub-problems on the distributed computing cores. We used an open-source nonlinear optimization (NLopt) (*Johnson, 2019*) library as the inner optimization solver to solve the local optimization sub-problems. The results in subplots (B), except for the 12-mean-degree graph, show a trend of slight decrease of the maximum of inner iterations while the ADMM iterations increase. Comparing the results of mean-degree graphs with regular graphs in subplots (B) shows that in most cases there is a large gap between the maximum number of iterations for the inner solver. This implies that for regular graphs the local optimization subproblems are solved in fewer iterations compared with those of mean-degree graphs. This is most visible for degree *d* = {4, 6, 12} in subplots 7B-9B, for degree 24 in subplot 10B, and for degree 96 in subplot 11B. For the ring graph the optimization sub-problems are solved with the same or fewer iterations than those of the

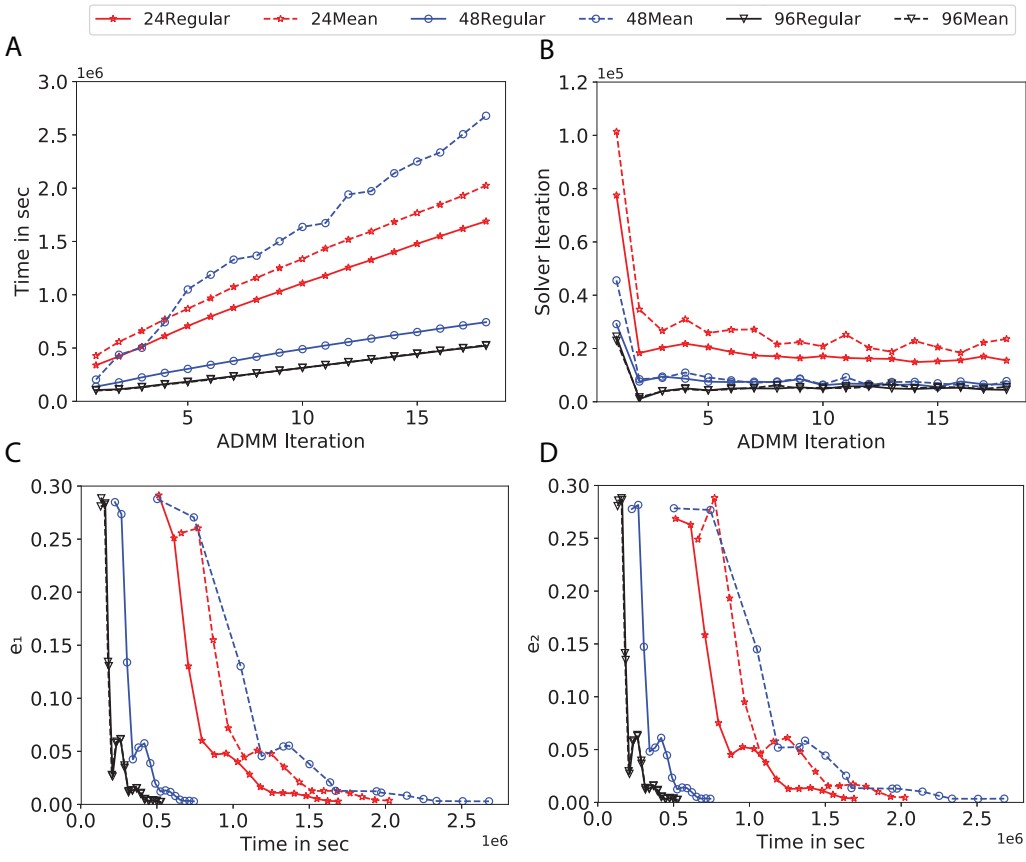

**Figure 10 Total time, maximum iterations of the inner optimization solver, and time vs. e1 and e2 for `Higgs` using $d$-mean-degree and $d$-regular graphs where $d = \{24, 48, 96\}$.** (A) Total elapsed time of the training phase, (B) Maximum iteration in the inner optimization solver, (C) Difference between the result of node j and its neighbors, time vs. e1 and (D) Difference between the result of node j and the average results, time vs. e2.

line graph. The results shown by subplots (B) might suggest that for regular graphs the consensus reaches faster than that of for mean-degree graphs which leads to a fewer number of inner optimization iterations. Besides, the maximum number of iterations for mean-degree graphs varies more than that of regular graphs. The maximum iterations regarding the 2-D torus graph stay very close to the maximum iterations regarding the 4-regular graph and that is better than that of the 4-mean-degree graph.

Subplots (C) in Figs. 7–11 show the $e_1$ as the training proceeds. Here, $e_1$ represents the difference between the parameters of the local and neighboring classifiers. The results shown by all the subplots (C) confirm that the error decreases as the training proceeds for all the graphs. Comparing the results of regular with mean-degree graphs shows a clear shift of the plots, i.e., regular graphs reach the lower errors faster than mean-degree graphs. This is most visible for all the degrees except for degree 3 in which the gap between the errors in the mean-degree and regular graphs are minor. The error for line graphs is similar or slightly less than that of the ring graphs, i.e., it seems that for the line graph the local classifiers are more similar to the neighboring classifiers compared to the ring graph.

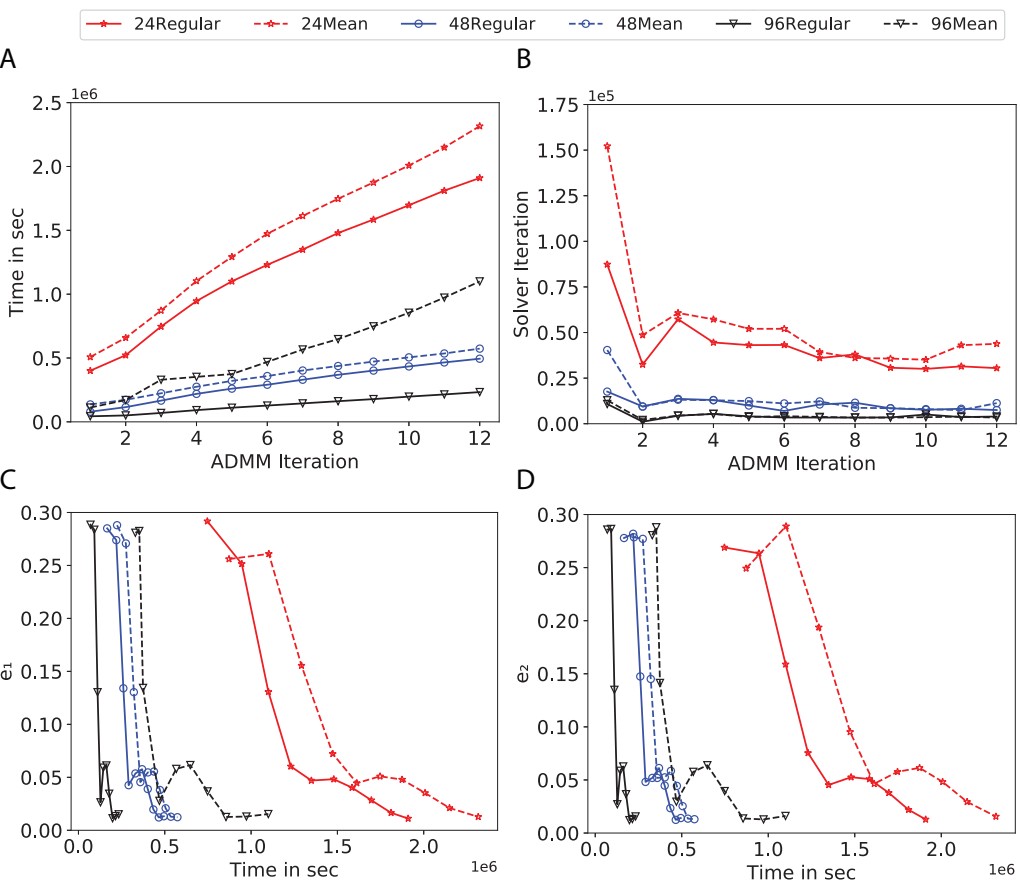

**Figure 11** **Total time, maximum iterations of the inner optimization solver, and time vs. e1 and e2 for** `Covtype` **using** $d$**-mean-degree and** $d$**-regular graphs where** $d = \{24, 48, 96\}$**.** (A) Total elapsed time of the training phase, (B) Maximum iteration in the inner optimization solver, (C) Difference between the result of node j and its neighbors, time vs. e1 and (D) Difference between the result of node j and the average results, time vs. e2.

This might suggest that nodes reach faster consensus with their neighbors when the number of neighbor nodes is few. For 2-D torus graphs results are as follows; in Fig. 7C, the error is less than that of the 4-regular graph until the 7th ADMM iteration and as the ADMM iteration proceeds the error gets worse than that of the 4-regular graph. In Figs. 8C and 9C, the error is slightly better than that of 4-regular graph. Note for degrees in which the gap between the results of regular and mean-degree graphs are not noticeable it turns out the connectivity of those graphs are similar.

Subplots (D) in Figs. 7–11 show the $e_2$ as the training proceeds. Here, $e_2$ represents the difference between the parameters of the local and global classifiers. To obtain the global classifier, we calculate the average of all the classifiers. The results shown by all the subplots (D) confirm that the error decreases as the training proceeds for all the graphs, except for the 2-D torus in Fig. 7D. Comparing the results of regular with mean-degree graphs shows a clear shift of the plots, i.e., regular graphs reach the lower errors faster than mean-degree graphs. The most notable trend is as follows; the gap between the errors in the regular and mean degree graphs is the largest for degree $d = \{3, 6, 12\}$ in

Figs. 7D and 8D, for degree $d = \{3, 4, 6\}$ in Fig. 9D, for degree 48 in Fig. 10D, and finally for degree 24 and 96 in Fig. 11D. For degrees in which the gap between the results of regular and mean-degree graphs are not noticeable it turns out the connectivity of those graphs are similar. Unlike in subplots (C), the results demonstrated in subplots (D) show that, except for Fig. 9D, the gap between the errors of the line and ring graphs is major, i.e., the difference between the local classifiers and the global classifier is large for line graphs compare to ring graphs. By comparing the results of subplots (C) and (D), we observe that although each node might reach faster consensus with its neighboring node using line graph compared to the ring graph, it might not reach the global consensus faster in that setting.

Comparisons of the results shown by all the subplots reveals that the magnitude of the gaps in subplots (A) are in line with the magnitude of the gaps in subplots (C) and (D) for the given degree. This is visible in Fig. 7 in which the large gaps between the results of the regular and mean-degree graphs in subplots (A), (C), and (D) belong to degree 6 and 12. Likewise, in most cases this is valid for the adverse cases, i.e., the smallest gaps between the results of regular and mean-degree graphs in subplots (A) matches of those in subplots (C) and (D) for the given degree. In Fig. 10, the smallest gap belongs to degree 24 and 96 in subplot (A) which is also the case in subpot (C) and (D). Comparison of subplots (C) and (D) shows that even if the gap between local classifiers and the neighboring classifiers is minor it does not mean that the local classifiers are close to the global classifier. This is notable for ring and line graphs.

In the next part of the experiments, we proceed the training using graphs with higher degrees, $d = \{100, 110, 120, 130, 140, 150, 160, 170, 190, 200\}$. It seems that the random mean-degree graphs have similar or not significantly different behavior than the random regular graphs for higher degrees since when the degrees become larger the mean degree graphs become very similar to the regular graph. The corresponding results are shown in Figs. 4 and 5 in the supplement. The results confirm that the number of iterations decreases as the degree of graphs increase even though there is no significant differences between regular and mean-degree graphs. Besides, the communication time increases as the graph degree increases this is due to reach the consensus with many nodes.

Finally, comparing the total training time between the low and high degree graphs reveals that the total elapsed time significantly increases for the lower degree graphs than that of the higher degree graphs given the fixed number of ADMM iterations.

We exclude all the results regarding the star graph from subplots (A)–(D) due to totally different scale of the results. For all the aforementioned metrics, namely total elapsed time, maximum iteration in the inner solver, $e_1$ and $e_2$, the star graph performed much worse than those of all the graphs constructed.

### Further error analysis

We study the difference between the results of each node $V_j$ and the average results of the neighboring nodes $V_i$, i.e., $|V_i - V_j|$. Figures 6 and 7 in the supplement show that increasing the degree leads to having smaller errors. This suggests that for higher degree graphs the result of each node gets closer to the results of its neighboring nodes even

though each node communicates with many nodes to reach the consensus for the given ADMM iteration. Boxplots in Figs. 6 and 7 in the supplement show a clear shift of decreasing error from left to right as the graph degree increases. This shift is visible in all the subplots. Although the median lines, the orange horizontal line in the body of boxplots, follow the shift, the decreasing trend eventually plateaus as the graphs gets closer to the complete graph. In line with the results in the previous sections, this might suggest that to get the error reduced, it is not necessary to use the complete graphs or the fully connected networks.

## CONCLUSION

The results from the previous section suggest that the performance of a distributed ADMM-based SVMs algorithm in terms of the number of iterations improves as the degree of graphs becomes larger, i.e., in the group of expander graphs with the fixed number of nodes, the graphs with higher connectivity exhibit accelerated convergence and the complete graphs outperform the expanders. Although this is expected, it turns out that increasing the connectivity of graphs all the way towards complete graphs in some cases increase the number of iterations. This can be explained with the increasing number of neighbors for each node and the need for each node to reach consensus with all neighbors, i.e., increasing the degree of graphs from 200 to 239 means that each node should reach the agreement with 239 instead of 200 neighbors, which, in turn increases the number of iterations needed and the communication time. This suggests that in order to achieve the sufficient level of convergence, it may not be needed to implement complete graphs as we used as our baseline and the expander graphs with lower connectivity will do. Replacing complete graphs as the underlying networks with expander graphs with lower degree is preferred in edge computing since lower degrees mean less communication of each node with neighbors which is preferred in the applications of edge computing with preserving privacy.

The next interesting point is that in the group of mean degree graphs, the second smallest eigenvalue of the Laplacian matrix representing the connectivity of the graph has higher variation compared to expander graphs. As shown in Fig. 3A random $d$-mean degree graph may have better connectivity than a random $d'$ mean degree graph even if $d < d'$, but this is not the case for random regular graphs. This describes the inconsistent performance of the algorithm using mean degree graphs in the Results section. Even though the difference in the degree distribution between the ring and line graphs is minor, the ring graphs outperform the line graphs. These findings support the notion that the homogeneity of degrees affects the performance of the algorithm. From this standpoint, how the nodes are connected is also influence the performance. This may be the reason why performance on a 2-D torus was worse compared to a 4-regular graph in some circumstances.

Another promising finding was that the diameter of mean degree graphs is roughly twice larger than those in random regular graphs. Note we used the same procedure of constructing random mean degree graphs as random regular graphs except the extra

conditions to obtain the expander graphs, i.e., the mean degree graphs used in the experiments were the base for generating random regular/expander graphs constructed. The comparisons of the graphs revealed that to reach the consensus, the communication between the nodes in the mean degree graphs is not as efficient as that in the regular graphs. This suggests that the expander graphs are a good candidate to be used in a network-based distributed algorithm.

## ACKNOWLEDGEMENTS

The parallel computations in this article are performed on Hebbe at Chalmers Centre for Computational Science and Engineering (C3SE) provided by the Swedish National Infrastructure for Computing (SNIC).

### Funding
The authors received no funding for this work.

### Competing Interests
Alexander Schliep is an Academic Editor for PeerJ.

### Author Contributions
- Shirin Tavara conceived and designed the experiments, performed the experiments, analyzed the data, performed the computation work, prepared figures and/or tables, authored or reviewed drafts of the paper, and approved the final draft.
- Alexander Schliep conceived and designed the experiments, analyzed the data, authored or reviewed drafts of the paper, and approved the final draft.

### Data Availability
Code available at https://github.com/GraduatePishi/EffectsOfNetworkTopologyADMM-SVMs-code.git

Data is available at the Schliep lab's Zenodo repository (https://zenodo.org/communities/schlieplab/?page=1&size=20):

- Shirin Tavara, & Alexander Schliep. (2020, December 30). Effects of Network Topology On the Performance of Consensus and Distributed Learning of SVMs Using ADMM. Zenodo. DOI 10.5281/zenodo.4406001.

- Wiedenhoeft, J., Brugel, E., & Schliep, A. (2016). HaMMLET - Supplemental Material [Data set]. Zenodo. DOI 10.5281/zenodo.46263.

- Lopes, I. de O. N. (2016). Automatic learning of pre-miRNAs from different species – Supplemental Material [Data set]. Zenodo. DOI 10.5281/zenodo.49754.

### Supplemental Information
Supplemental information for this article can be found online at http://dx.doi.org/10.7717/peerj-cs.397#supplemental-information.

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
