# Peer review of "Effects of network topology on the performance of consensus and distributed learning of SVMs using ADMM"

_PeerJ Computer Science, doi:10.7717/peerj-cs.397_

## Round 0.1 · original submission · Minor Revisions

This paper was reviewed by two experts. Both of them raised significant concerns about how ADMM is integrated with SVM. The authors are thus suggested to put efforts to explain the transformation/integration process.

Reviewer 1 ·

Basic reporting

.

Experimental design

.

Validity of the findings

.

Additional comments

This paper investigated the key features of networks for distributed ADMM in the applications of SVMs. Extensive experiments have been conducted for this purpose. However, there are still several problems need to be addressed.
1. As the ADMM is the key method used in this paper, it should appear in the title.
2. The section ADMM is confusing. The problems (12) and (13) are not formal. The process of the used consensus-based ADMM algorithm should be provided. It will be better if the convergence analysis could be given rather than only presenting the experiment stduies.
3. The transformation from SVM to ADMM is absent.

Reviewer 2 ·

Basic reporting

The wording of this article is more accurate, in line with the requirements of politeness and norms.This paper introduces the research background and current situation of the problem.The graphs are also clear.

Experimental design

This article should further explain how the Alternating Direction Method Of Multipliers (ADMM) algorithm can be applied to Support Vector Machine (SVM) training.Instead of just describing the Alternating Direction Method Of Multipliers algorithm and Support Vector Machine training model respectively.The article needs to add more details about the transformation between models.

Validity of the findings

(1)The simulation results in this paper are more comprehensive, and the conclusion of this paper is validated effectively.
(2)The computational part is persuasive.The parallel computations in this article are performed on Hebbe at Chalmers Centre for Computational 485 Science and Engineering (C3SE) provided by the Swedish National Infrastructure for Computing (SNIC).

Additional comments

(1)The paper should organize the references more strictly and unify the quotation format of multiple references.
(2)The language needs to pay more attention to details, such as the use of parentheses when referring to graphs.

---

## Round 0.2 · accepted · Accept

The concerns raised by the reviewers have been addressed, so the paper can be accepted for publication in this journal. Congratulations to the authors!

Reviewer 1 ·

Basic reporting

N/A

Experimental design

N/A

Validity of the findings

N/A

Additional comments

The author has addressed most of my comments. I have no other concerns.

Reviewer 2 ·

Basic reporting

No comment.

Experimental design

No comment.

Validity of the findings

No comment.

Additional comments

This paper investigates the impact of network topology on the performance of an ADMM-based learning of Support Vector Machine (SVM) , and mean-degree graphs, and some common modern network topologies. Further, the paper investigates to which degree the expansion property of the network influences convergence for iterations, training and communication time. The experimental part is very sufficient. I have no doubts now.